# An improved control strategy of power-frequency characteristics of virtual synchronous generator

Weiqiang Zheng[1], Modan Su[1], Changwei Gao[2]*

**1** Yingkou Power Supply Company, Yingkou, China, **2** Liaoning Institute of Science and Technology, Benxi, China

* gaochangwei2008@163.com

## Abstract

A coupling relationship exists between virtual inertia $J$ and damping coefficient $D$ in traditional virtual synchronous generator (VSG). While a large virtual inertia enhances the VSG's frequency support capability, it may also induce active power oscillations or significant overshoot during load power variations. To address this issue, the paper proposes an improved power–frequency characteristic control strategy for VSG. First, the mathematical model of the conventional VSG is analyzed, and a closed-loop transfer function capturing the dynamic relationship between active power and frequency fluctuation is established. The influences of virtual inertia and damping on the active power response during transient processes are systematically investigated. Second, a leading differential element is incorporated into the traditional VSG power–frequency controller, yielding an enhanced control strategy for regulating VSG power–frequency characteristics. The resulting power–frequency behavior is rigorously analyzed under both off-grid and grid-connected operating conditions. Experimental results demonstrate that, by appropriately tuning the parameters of the lead-differential element, active power oscillations during transient processes can be effectively suppressed. Moreover, the proposed strategy achieves effective decoupling between dynamic response characteristics and steady-state performance.

## Introduction

In recent years, to mitigate carbon emissions, new energy power generation represented by wind energy and solar energy has experienced rapid development [1,2]. Such generation systems are integrated into the power grid via power electronic converters. While offering advantages such as flexible control and fast dynamic response, they inherently lack the rotational inertia characteristic of conventional synchronous generator (SG), thereby compromising system stability [3]. With the deployment and implementation of the "carbon peak and carbon neutrality" objective, the penetration rate of new energy in the power grid is expected to rise further, bringing substantial challenges to grid stability [4]. To facilitate the grid-friendly integration

**Data availability statement:** All relevant data are within the manuscript.

**Funding:** This study was funded by science and technology project of Liaoning Electric Power Co., Ltd. "Research on optimal control method of power frequency active support capability of PV virtual synchronous power supply (2025YF-34)" The funders had no role in study design, data collection and analysis, decision to publish, or preparation of the manuscript.

of new energy and improve the operation stability of new energy power generation, scholars have proposed virtual synchronous generator (VSG) control theory.

On the basis of simulating the external droop characteristics of primary frequency modulation and primary voltage regulation of SG, VSG further simulates the rotor motion characteristics of SG. It constructs a rotor motion equation in the power-frequency control loop, thereby enabling the VSG to exhibit inertia [5,6]. While simulating the rotor inertia of SG enhances the frequency stability of the VSG, it also introduces the issue of power oscillation [7–9]. The introduction of inertia converts the originally first-order characteristics of the active power control loop into typical second-order oscillatory characteristics, which can cause significant transient power oscillations in the VSG when the grid is subjected to external disturbances [10–12]. At present, the improvement of VSG power-frequency control strategy primarily focus on the flexible configuration of VSG parameters. Optimization algorithms are employed to calculate and adjust its virtual inertia and damping in real time, thereby enabling the VSG system to achieve better dynamic performance and suppress system oscillations [13]. Increasing the damping coefficient in the rotor motion equation can well suppress this oscillation [14,15]. However, the damping coefficient and the droop coefficient in VSG control are essentially the same coefficient, increasing the damping also raises the droop coefficient [16–18]. According to the characteristics of primary frequency modulation, a larger droop coefficient leads to a greater steady-state deviation in the VSG output power when grid frequency deviates. Moreover, while a larger virtual inertia in VSG improves system frequency stability, it also results in more pronounced output power oscillations, larger overshoot, and longer settling times [19–21].Therefore, there exists a certain contradiction between the dynamic and steady-state performance of the VSG, and its primary frequency regulation characteristics are coupled with its damping characteristics. It is difficult to solve the problem by optimizing the control parameters of VSG, which brings difficult problems to the suppression of VSG power oscillation.

To address the system power-frequency oscillation problem caused by the coupling between the dynamic characteristics and steady-state performance of the VSG, Reference [22] independently designs two control links, primary frequency modulation and damping, which effectively weakens the coupling between VSG damping parameters and frequency modulation coefficients. However, there is still a risk of low-frequency oscillation when the system is subjected to significant disturbances. Reference [23] designs state feedback variables based on modern control theory, which improves the dynamic characteristics of the system during transient process. However, the improvement in steady-state performance is not substantial. References [24,25] attempt to construct the objective function includes frequency deviation and frequency change rate, employing a fuzzy control algorithm to comprehensively adjust damping and inertia during transient states. Although the control performance is satisfactory, but the design of the control rule base requires extensive sample data, making practical implementation cumbersome. Reference [26] introduces an additional degree of freedom through fractional-order control, which further improves the inertial response and achieves achieves effective

oscillation suppression. However, the design of fractional-order is cumbersome, and the fractional stability analysis is difficult, so it is difficult to analyze the actual physical meaning of the design. Reference [27] proposes a control strategy that dynamically adjusts virtual inertia using a radial basis function (RBF) neural network. However, it restricts the dynamic adjustment range of damping coefficient and lacks adjustment flexibility. Reference [28] combines virtual inertia with fractional-order, proposing an adaptive control strategy that incorporates virtual inertia, damping coefficient, and dynamic damping compensation. But the accuracy of dynamic adjustment of system damping in response to frequency variations requires further improvement. In Reference [29], power coupling is attributed to variations in output voltage. Based on the virtual impedance method, a voltage compensation term is constructed to achieve power decoupling under high resistance-inductance ratio and different operating conditions. However, disturbances can still lead to dynamic oscillations and potential steady-state errors. Reference [30] introduces active and reactive power interaction control in droop control and uses the relative gain array (RGA) to analyze system coupling. However, the parameter selection in this work is complex and requires reconfiguration for different operating conditions. Reference [31] incorporates active power as a feed-forward term to adjust the output voltage reference, achieving power decoupling through voltage compensation. However, the design of high-order decoupling controllers is challenging and could benefit from simplification.

To enhance the control flexibility of the VSG and achieve decoupled control of its dynamic characteristics and steady-state performance, this paper proposes a novel improved VSG power-frequency characteristic control strategy. First, the power-frequency characteristics of traditional VSG are analyzed under both off-grid and grid-connected operating conditions, clarifying the coupling mechanism between its dynamic response and steady-state performance. Second, the VSG active power oscillation mechanism is analyzed using the power angle characteristic curve of the synchronous generator. Based on this analysis, a leading differential control loop is introduced into the traditional VSG power frequency controller, combined with an inertia compensation mechanism, to eliminate the inherent parameter coupling. This approach achieves decoupled control of the system's dynamic and steady-state performance.

## Traditional VSG power-frequency characteristics

### Power-frequency characteristics of traditional VSG in off-grid operation

Based on the second-order electromechanical transient equation of synchronous generator (SG), the mathematical model of the traditional VSG is described by equations (1) and (2).

$$\begin{cases} J\frac{\mathrm{d}\Delta\omega}{\mathrm{d}t} = \frac{P_\mathrm{m}}{\omega_0} - \frac{P_\mathrm{e}}{\omega_0} - D \cdot \Delta\omega \\ \frac{\mathrm{d}\theta}{\mathrm{d}t} = \omega \end{cases} \tag{1}$$

$$\dot{E}_0 = \dot{U} + \dot{I}\,(R + jX) \tag{2}$$

Where $J$ is the virtual inertia, $P_\mathrm{m}$ and $P_\mathrm{e}$ are virtual mechanical power and virtual electromagnetic power respectively, $D$ is the damping coefficient, $\Delta\omega$ is the deviation between the actual angular frequency and the synchronous angular frequency ($\Delta\omega = \omega - \omega_0$), $E_0$ is the excitation electromotive force, $U$ and $I$ are the virtual armature voltage and current, $R$ and $X$ are the virtual armature resistance and synchronous reactance; $\theta$ is the virtual electrical angle.

Based on the design theory of SG active power and frequency control system, the power frequency control unit of VSG consists of rotor motion equation and *P-f* droop control. The virtual mechanical power $P_\mathrm{m}$ of VSG is derived from the deviation $\Delta P$ between the active power reference value $P_\mathrm{ref}$ and the actual output power $P$, and the mathematical model of the *P-f* droop is given by equation (3).

$$P_m = P_{ref} + \Delta P = P_{ref} + K_P(f_{ref} - f) \qquad (3)$$

Where $K_p$ is the frequency modulation coefficient, $f_{ref}$ is the frequency reference value, and $f$ is the actual frequency.

During off-grid operation, the frequency of the VSG varies with load fluctuations. The droop characteristic is employed to adjust the active power output, thereby maintaining active power balance. The P-f droop characteristic given in equation (3) can be converted to P-ω droop characteristic shown in equation (4).

$$P_m = P_{ref} + \frac{K_P}{2\pi}(\omega_0 - \omega) = P_{ref} + D_P(\omega_0 - \omega) \qquad (4)$$

Where $D_p = K_p/2\pi$.

Substituting equation (4) into equation (1) yields the transfer function between the VSG frequency deviation and the active power deviation, as expressed in equation (5)

$$G_1(s) = \frac{\Delta\omega(s)}{\Delta P(s)} = \frac{\omega - \omega_0}{P_e - P_{ref}} = -\frac{1}{J\omega_0 s + D\omega_0 + D_P} \qquad (5)$$

The damping coefficient $D$ and the droop coefficient $D_p$ are essentially frequency droop adjustment coefficients, and they perform equivalent functions. Therefore, they can be consolidated into a single equivalent damping coefficient, denoted as $D$. As a result, equation (5) can be simplified to equation (6).

$$G1(s) = \frac{\Delta\omega(s)}{\Delta P(s)} = \frac{\omega - \omega 0}{Pe - Pref} = -\frac{1}{J\omega 0 s + D\omega 0} = -\frac{K1}{Ts + 1} \qquad (6)$$

$$\begin{cases} K_1 = \frac{1}{D\omega_0} \\ T = \frac{J}{D} \end{cases} \qquad (7)$$

As shown in equation (6), when the VSG operates in off-grid mode, its frequency response exhibits first-order inertial characteristics. $K_1$ is the first-order inertia coefficient, whose value is related to the damping $D$. $T$ is the first-order inertia time constant, which is determined by virtual inertia $J$ and damping coefficient $D$.

Fig 1 shows the frequency step response of VSG when the load power increases under the action of the same damping coefficient $D$ but with different virtual inertia $J$ values. Its power frequency control exhibits a droop characteristic with distinct inertial properties, which not only adjust the frequency in response to load variations but also effectively prevents the excessive frequency fluctuation. It can be observed from Fig 1, when $D$ is constant, an increase in $J$ leads to a smoother frequency change trajectory and a significantly longer response time, whereas the steady-state frequency deviation remains unchanged. The steady-state frequency deviation corresponding to different virtual inertia $J$ values is consistently 0.2 Hz.

Fig 2 shows the frequency step response of VSG when $J$ is constant and $D$ is different, and the load power increases. When the virtual inertia $J$ is constant, an increase in the damping coefficient $D$ leads to a reduction in both the frequency response time and the steady-state frequency deviation. As $D$ increases from 1 to 4, the steady-state frequency deviation of the system decreases from 0.2 Hz to 0.05 Hz. This result indicates that the virtual inertia $J$ primarily influences the dynamic behavior of the frequency response, whereas the damping coefficient $D$ affects not only the dynamic characteristics but also the steady-state deviation of the frequency.

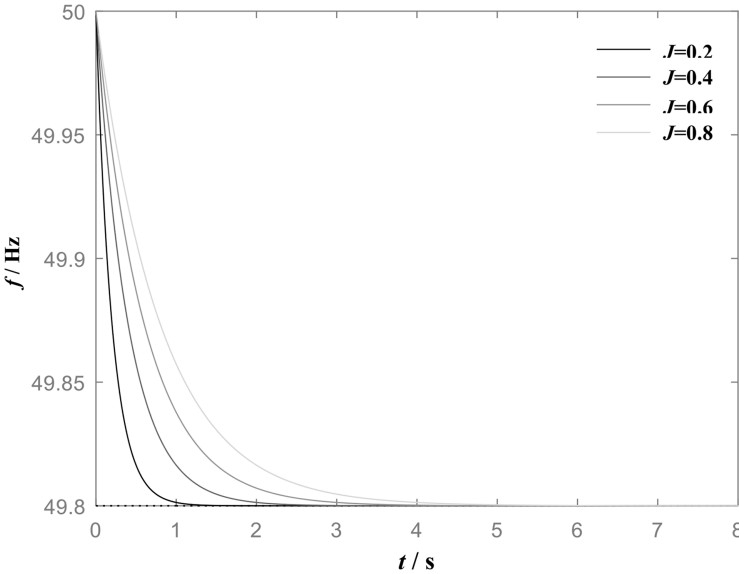

**Fig 1. Frequency step response of VSG when *D* is constant (*D*=1) and *J* is different.**

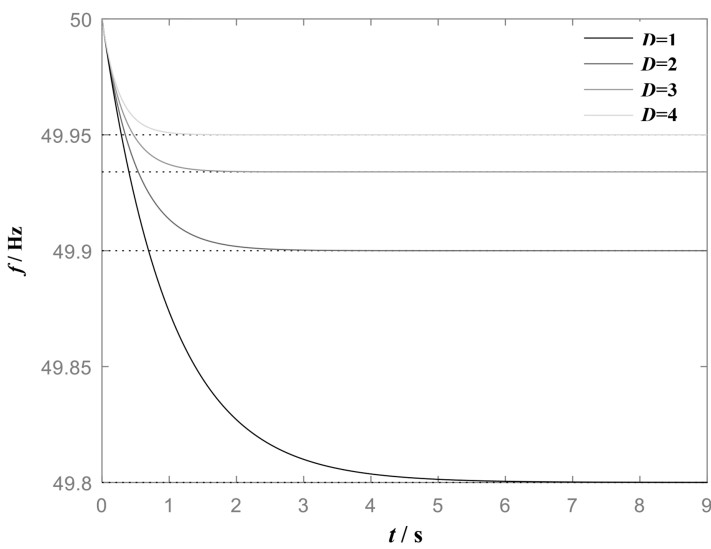

**Fig 2. Frequency step response of VSG when J is constant(J=1 kg•m2) and D is different.**

## Power-frequency characteristics of traditional VSG in on-grid operation

When VSG is connected to grid, its output voltage and frequency are restricted by the grid, while its output power is controlled according to the grid-connected dispatch demand. The output power of the grid-connected VSG is given by equation (8).

$$\begin{cases} P_e = \dfrac{EU_s}{Z}\cos(\alpha - \delta) - \dfrac{U_s^2}{Z}\cos\alpha \\ Q_e = \dfrac{EU_s}{Z}\sin(\alpha - \delta) - \dfrac{U_s^2}{Z}\sin\alpha \end{cases} \tag{8}$$

$E$ and $U_\text{S}$ represent the voltage of the VSG and the grid voltage. $\delta$ is the phase difference between them. $\alpha$ is the impedance angle, and $Z = R + jX$ represents the total equivalent impedance, which is the sum of the inverter's output impedance and the line impedance. When the line resistance is ignored, equation (9) is obtained.

$$P_\text{e} = \frac{EU_\text{S}}{X}\sin\delta \tag{9}$$

Usually, $\delta$ is very small. Therefore, equation (9) can be simplified as equation (10).

$$P_\text{e} = \frac{EU_\text{S}}{X}\delta = K\delta \tag{10}$$

$$\delta = \int (\omega - \omega 0)\text{d}t \tag{11}$$

When VSG is connected to grid, its closed-loop power control structure is shown in Fig 3, $K_\omega$ is the active power frequency droop coefficient, and $\omega_\text{g}$ is the grid angular frequency.

For infinite capacity power system, $\omega_\text{g} = \omega_0$, the closed-loop transfer function for the active power response of the grid-connected VSG is given by Equation (12).

$$G_2\left(s\right) = \frac{EU_\text{S}/J\omega_0 X}{s^2 + (D/J)s + EU_\text{S}/J\omega_0 X} \tag{12}$$

The system described by equation (12) is a typical second-order system. Its undamped natural oscillation angular frequency and damping are described as equation (13). As shown in Equation (13), the system's natural frequency is determined solely by the virtual inertia $J$, and an increase in $J$ reduces this frequency. The damping ratio of the system is related to both virtual inertia $J$ and damping $D$. Reducing $J$ or increasing $D$ enhances the damping ratio, thereby helping to mitigate system power overshoot.

Based on equation (12), when damping coefficient $D$ held constant and inertia $J$ set to different values, the active power step response of the grid-connected VSG is shown in Fig 4.

It can be seen that, with constant damping $D$, a larger $J$ leads to a more pronounced overshoot in active power and a longer dynamic adjustment time.

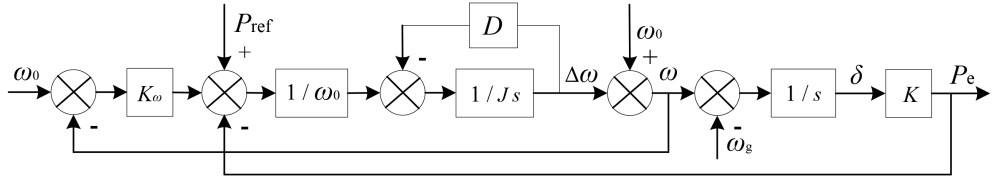

**Fig 3. The power closed-loop control structure of VSG.**

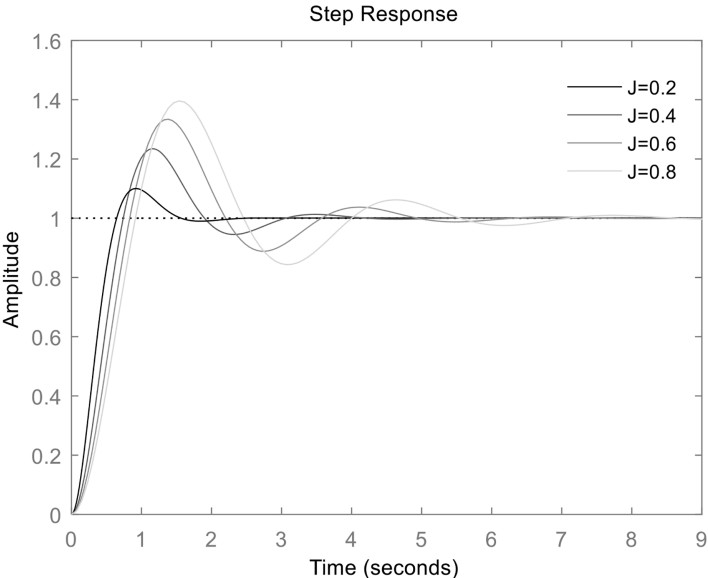

**Fig 4. Frequency step response of VSG when $D$ is constant ($D=1$) and $J$ is different.**

$$\begin{cases} \omega_{2n} = \sqrt{\dfrac{EU_s}{J\omega_0 X}} \\ \zeta_2 = \dfrac{D}{2}\sqrt{\dfrac{\omega_0 X_s}{JEU}} \end{cases}$$

(13)

Fig 5 presents the active power step response of the grid-connected VSG for a constant virtual inertia $J$ and different damping coefficients D. It can be observed that, for a fixed $J$, increasing $D$ leads to a reduction in both the oscillation amplitude and frequency, as well as a shorter dynamic adjustment time.

The distribution of the closed-loop poles of $G_2(s)$ for different virtual inertia ($J=1,3,10$) and damping D varies from zero to infinity is shown in Fig 6. The transfer function $G_2(s)$ possesses two characteristic roots, $s_1$ and $s_2$. When $J$ is small, both roots always lie on the real axis. $s_1$ is the non-dominant pole, located far from the imaginary axis, whereas $s_2$ is the dominant pole, situated closer to it. As $D$ increases gradually, $s_1$ moves away from the imaginary axis, while $s_2$ approaches it, causing $G_2(s)$ to approximate a first-order system.

When the virtual inertia $J$ is large, the two characteristic roots form a pair of conjugate complex numbers. As the damping $D$ increases, $s_1$ and $s_2$ move in the direction indicated by the arrows. When $D$ increases to a certain value, the roots eventually become two real poles, and the system changes from under-damped operation state to over-damped operation state. A larger $J$ brings the characteristic roots closer to the imaginary axis, leading to a longer dynamic adjustment time and deteriorated stability.

If $D$ is held constant, a larger $J$ weakens the damping effect, resulting in stronger power-frequency oscillations and reduced transient stability in the VSG. If $J$ is fixed, a larger $D$ strengthens system damping, which is effective in suppressing oscillations. Nevertheless, the dynamic response speed of the VSG system slows down, and the adjustment time becomes longer. Furthermore, excessive damping can make the dominant pole excessively close to the imaginary axis, ultimately compromising system stability.

Due to the coupling between VSG's virtual inertia $J$ and damping $D$, there is a contradiction between its dynamic regulation characteristics and steady-state performance. Decoupling control of the $J$ and $D$ enhance the control freedom, thereby improving both dynamic response and steady-state performance of the system.

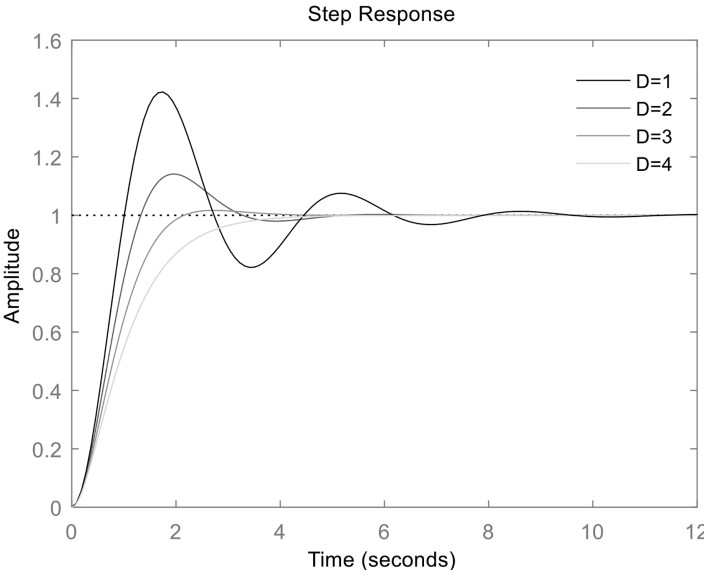

**Fig 5. Frequency step response of VSG when *J* is constant(*J*=1 kg•m²) and *D* is different.**

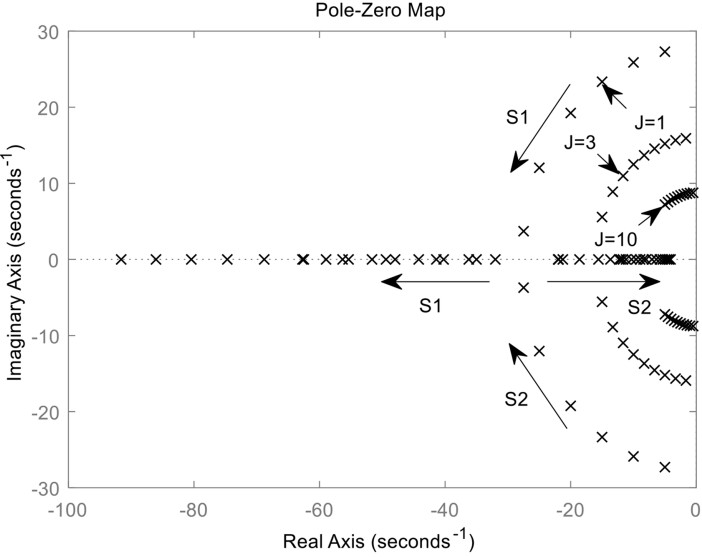

**Fig 6. Closed-loop pole distribution of $G_2(s)$.**

### VSG power-frequency characteristics control strategy based on leading differential control

**Principle of leading differential control.** In automatic control systems, a differential control link can be introduced to achieve advanced correction of system signals, thereby increasing system damping and accelerating the response speed without compromising steady-state control accuracy. Therefore, this paper proposes to incorporate a lead differential control link into the VSG power frequency controller and perform inertia correction to improve the control performance.

There are two modes for introducing the leading differential control link into the power frequency controller. One is the the damped inner-loop leading differential control, denoted as position A in Fig 7. The other is the damped outer loop leading differential control, marked as position B in Fig 7. The power-frequency characteristics of VSG under two control modes are analyzed below.

## Power-frequency characteristics of the improved VSG under off-grid operation condition

When the damped inner loop leading differential control is applied, it can be seen from Fig 6 that the transfer function relating VSG angular frequency to power deviation is shown in equation (14).

$$G_{A1}(s) = \frac{1 + Cs}{(J\omega_0 + CD\omega_0)s + D\omega_0} \tag{14}$$

Where, $C$ is the adjustment parameter of the leading differential link. For a small C, $G_{A1}(s)$ exhibits a first-order inertia response. For a large C, $G_{A1}(s)$ presents a linear droop characteristic, whose droop coefficient is influenced by Damping $D$.

When the damped outer loop leading differential control is introduced, the transfer function relating the VSG angular frequency to the power deviation is shown in equation (15).

$$G_{B1}(s) = \frac{1 + Cs}{J\omega_0 s + D\omega_0} \tag{15}$$

As shown in Equation (15), the adjustment parameter $C$ affects the zero locations but not the pole positions.

After introducing the leading differential control link, the power-frequency characteristic of VSG gains an additional zero point, and the first-order inertia time constant of $G_{A1}(s)$ increases. The steady-state frequency deviation of the system still depends on the damping $D$, whereas adjusting the value of $C$ allows the zero and pole locations to be adjusted, thereby optimizing the dynamic characteristics of the system. Fig 8 and Fig 9 show the frequency response characteristics of $G_{A1}(s)$ and $G_{B1}(s)$ when $J$ and $D$ are constant and $C$ is different.

When the leading differential control link is introduced, both $G_{A1}(s)$ and $G_{B1}(s)$ exhibit droop characteristics during the initial stage of power step response, while both display inertial behavior in the later stage. The system therefore combines the fast-response attributes of droop control with the high inertia of a first-order element. Note that the value of $C$ does not affect the steady-state frequency deviation of the system. A comparison reveals that, for the same value of C, $G_{A1}(s)$ possesses greater inertia than $G_{B1}(s)$, whereas the droop characteristic of $G_{B1}(s)$ is more pronounced.

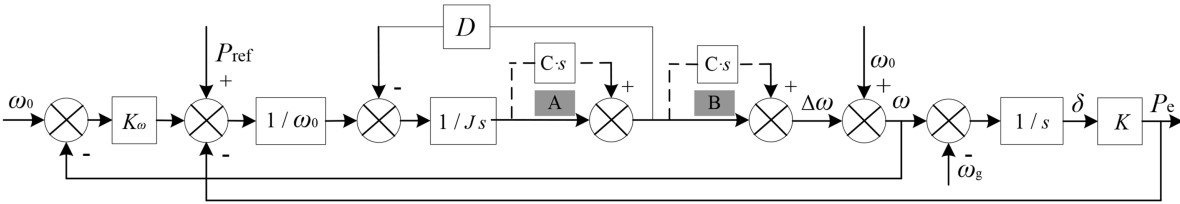

**Fig 7. VSG power-frequency control structure after introducing leading differential control link.**

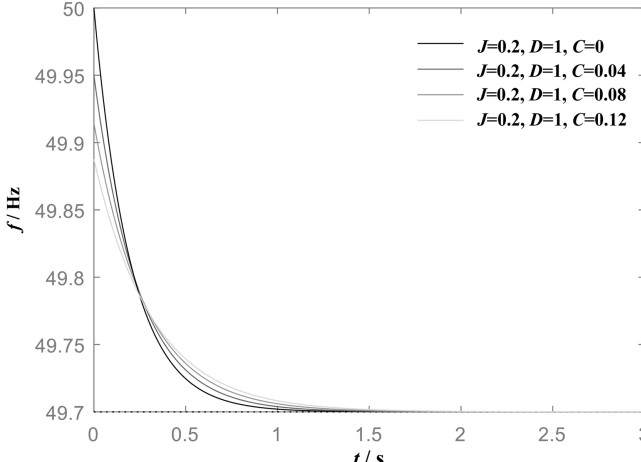

**Fig 8. Frequency response of the improved VSG based on leading differential control $G_{A1}(s)$.**

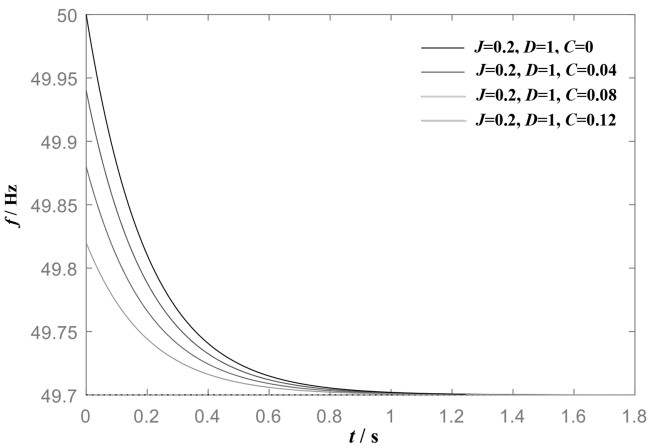

**Fig 9. Frequency response of the improved VSG based on leading differential control $G_{B1}(s)$.**

## Power-frequency characteristics of the improved VSG under on-grid operation condition

When the damped inner loop leading differential control is introduced, the improved VSG active power closed-loop transfer function is shown as equation (16).

$$G_{A2}(s) = \frac{K(1 + Cs)}{(J\omega_0 + CD\omega_0)s^2 + (D\omega_0 + KC)s + K}$$

(16)

When the damped outer loop leading differential control is introduced, the improved VSG active power closed-loop transfer function is shown as equation (17).

$$G_{B2}(s) = \frac{K(1 + Cs)}{J\omega_0 s^2 + (D\omega_0 + KC)s + K}$$

(17)

As shown in equations (16) and (17), both $G_{A2}(s)$ and $G_{B2}(s)$ are second-order systems with zeros, where the zero locations are determined by $C$. The natural oscillation frequencies $\omega_{n\_A2}$ and $\omega_{n\_B2}$ and the damping ratios $\xi_{A2}$ and $\xi_{B2}$ are given in equations (18) and (19), respectively.

Comparing with equation (14), the natural oscillation angular frequency relationship and damping ratio relationship of the VSG grid-connected system under the three control strategies are summarized in equation (20).

$$\begin{cases} \omega_{n\_A2} = \sqrt{\frac{K}{(J+CD)\omega_0}} \\ \zeta_{A2} = \frac{D\omega_0 + CK}{2\sqrt{(J+CD)K\omega_0}} \end{cases}$$

(18)

$$\begin{cases} \omega_{n\_B2} = \sqrt{\frac{K}{J\omega_0}} \\ \zeta_{B2} = \frac{D\omega_0 + CK}{2\sqrt{JK\omega_0}} \end{cases}$$

(19)

$$\begin{cases} \omega_{n\_B2} = \omega_{2n} > \omega_{n\_A2} \\ \zeta_{B2} > \zeta_2, \zeta_{B2} > \zeta_{A2} \end{cases}$$

(20)

$\xi_{B2}$ is always greater than $\xi_2$, and $\xi_{A2}$ exceeds $\xi_2$ within a certain range. In summary, the improved VSG power-frequency characteristic control strategy based on leading differential control enhances system damping and contributes to suppressing power overshoot. The power-frequency regulation characteristics of the system are as follows.

(1) The leading differential control has no effect on the power-frequency steady-state performance of VSG system, and the active power deviation depends solely on the damping coefficient $D$.

(2) The dynamic characteristics of the improved VSG based on leading differential control are determined by $J$, $D$ and $C$. By adjusting the parameter $C$ in the leading differential link, the closed-loop pole and zero locations can be modified, thereby facilitating the decoupling control of power-frequency dynamic characteristics and steady-state performance of the system.

(3) The leading differential control increases the damping ratio of the system and improves the dynamic characteristics of the system. In particular, the damping ratio of the VSG system improved with the damped outer-loop lead differential control is higher.

Based on the comparative analysis above, this paper improves the traditional VSG power-frequency controller by adopting damped outer loop leading differential control.

### Inertial correction of leading differential control unit

Based on the above analysis, it can be concluded that the introduction of leading differential control causes the power-frequency characteristics of the VSG to exhibit droop behavior during the initial stage of the dynamic response, which reduces the system inertia and leads to an excessive frequency drop.

Therefore, the inertia correction links can be added in series based on leading differential control, thereby compensating for the reduction in system inertia after the differential control is applied. Following the addition of the inertia correction

link, the proposed improved VSG power frequency control structure based on leading differential control is shown in Fig 10, where $T$ is the time constant of the inertial correction link.

**Power-frequency characteristics of the improved VSG under off-grid operation condition.** The transfer function of angular frequency change to power deviation of improved VSG system based on leading differential control is reformulated as equation (21).

$$G_{B1\_n}(s) = \frac{1 + Cs}{(J\omega_0 s + D\omega_0)(Ts + 1)}$$

(21)

$G_{B1\_n}$ has two pole points $(-1/T, -D/J)$, and one zero point $(-1/C)$. When $D$ and $J$ are constant, the positions of zero point and pole point can be configured by adjusting the values of $C$ and $T$, thereby optimizing the dynamic characteristics of $G_{B1\_n}$. The step response of $G_{B1\_n}$ for different values of $C$ and $T$ is shown in Fig 11.

When $D$ and $J$ are constant, the dynamic characteristic of $G_{B1\_n}$ is determined by $C$ and $T$. A comparison of Fig 11 with Fig 9 and Fig 1 shows that the frequency response of $G_{B1\_n}(s)$ lies between that of $G_1$ and $G_{B1}(s)$. Its inertia is greater than that of $G_{B1}(s)$ yet smaller than that of $G_1(s)$, while dynamic response speed is higher than that of $G_1(s)$ but lower than that of $G_{B1}(s)$. It can thus be concluded that, with $D$ and $J$ held constant, appropriate adjustment of $C$ and $T$ enables $G_{B1\_n}(s)$ to achieve both higher inertia and faster response.

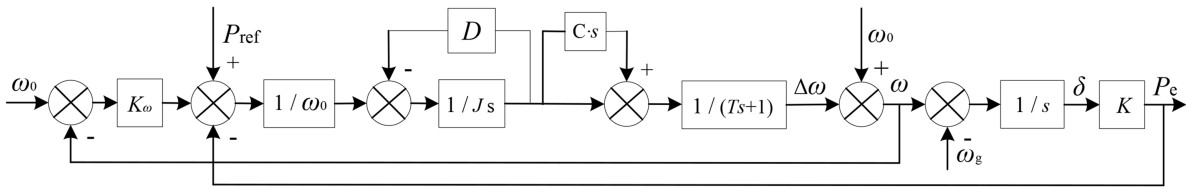

**Fig 10. Power-frequency control structure of the improved VSG based on leading differential control.**

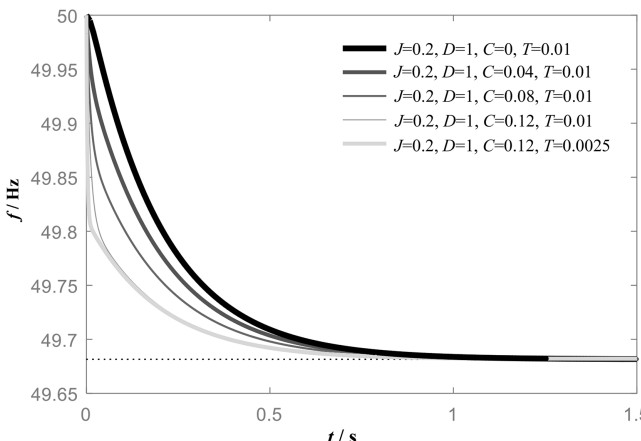

**Fig 11. Improved VSG frequency step response.**

  

**Power-frequency characteristics of the improved VSG under grid-connected operation condition.** In grid-connected mode, the closed-loop transfer function for the system active power is given by equation (22):

$$G_{B2\_n}(s) = \frac{K(1 + Cs)}{TJ\omega_0 s^3 + (TD\omega_0 + J\omega_0 + TCK)s^2 + (D\omega_0 + CK)s + K}$$

(22)

$G_{B2\_n}(s)$ is a typical third-order system transfer function, its damping ratio and natural oscillation frequency are not easy to calculate. In this paper, the influence of $C$ and $T$ on the system's dynamic behavior is examined by analyzing how the zeros and poles of $G_{B2\_n}(s)$ vary with $C$ and $T$.

Assuming $C=0$ and the value range of $T$ is 0~0.02, the zero and pole distribution of $G_{B2\_n}(s)$ is shown in Fig 12. When $C=0$, $G_{B2\_n}(s)$ has three poles: $s_1$ is a negative real pole, $s_2$ and $s_3$ are a pair of conjugate complex poles located in the left plane. When $T$ increases, all three poles move closer to the imaginary axis, resulting in a decrease in the system damping ratio, a slower dynamic response, and increased oscillation in the active power.

Assuming $T=0$ and letting $C$ vary from 0 to 0.02, the zero and pole distribution of $G_{B2\_n}(s)$ is shown in Fig 13. It can be seen that when $T=0$, $G_{B2\_n}(s)$ possesses one zero point z and a pair of conjugate complex poles $s_1$ and $s_2$, all located in the left half plane. As $C$ increases, the zero point gradually moves toward the imaginary axis, while the two poles gradually move away from it, thereby raising the system damping ratio, speeding up the dynamic response, and suppressing active-power oscillation. When $C$ reaches a certain value, the two poles meet on the real axis. Subsequently, $s_1$ moves away from the imaginary axis while $s_2$ moves toward it, with $s_2$ becoming the dominant pole.

Based on the analysis above, the following conclusions can be drawn:

(1) After the introduction of leading differential control, the added zeros and poles do not influence the steady-state performance of the system. The steady-state active power deviation depends solely on the damping coefficient $D$.

(2) The dynamic characteristics of the system are affected by $J$, $D$, $C$ and $T$, and the degrees of freedom for controlling the dynamic response are significantly enhanced.

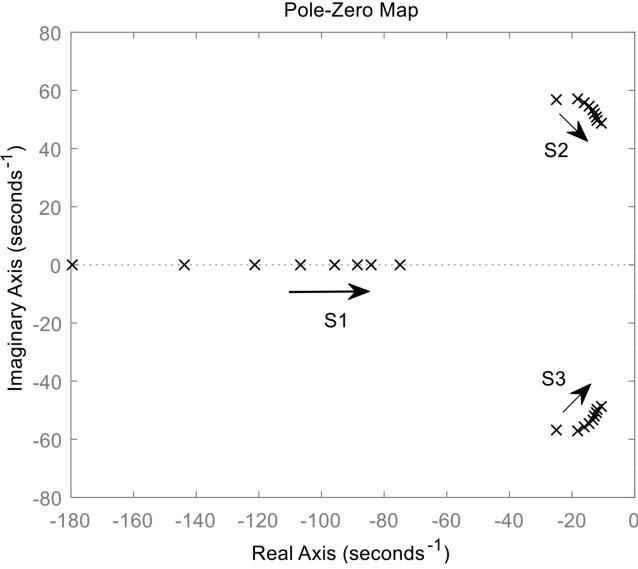

**Fig 12. The change trajectory of $G_{B2\_n}(s)$ pole when $T$ is changed.**

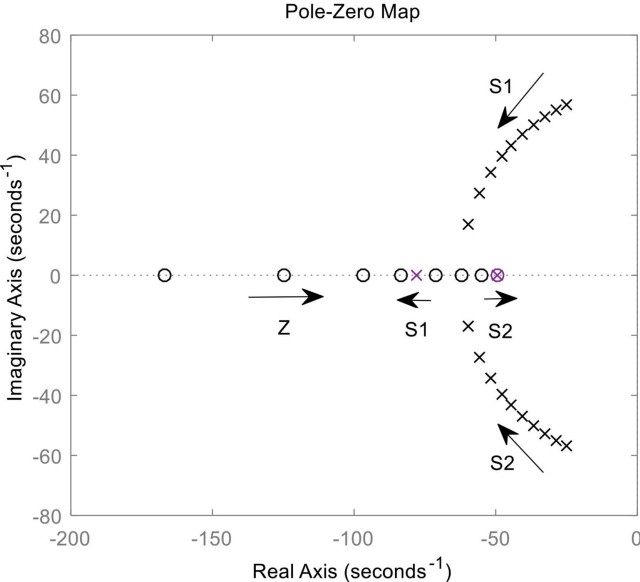

**Fig 13.  The change trajectory of $G_{B2\_n}$ (s) pole when $C$ is changed.**

(3) As $C$ increases, the damping ratio of the system rises, the dynamic response becomes faster, and power oscillation is suppressed. As $T$ increases, the damping ratio decreases, the dynamic response slows down, and power overshoot grows. By properly adjusting $C$ and $T$, the dynamic performance of the system can be effectively improved.

## Experiment verification

To verify the correctness and effectiveness of the improved VSG power-frequency characteristic control strategy based on leading differential control proposed in this paper, a test system for VSG off-grid and grid-connected operation is constructed, the topology of which is shown in Fig 14.

The experimental platform is shown in Fig 15, which is mainly composed of host computer, RT-LAB real-time simulator (HBUREP-100), DSP controller (TMS320F28335), oscilloscope and other components. The real-time simulator processor has a main frequency of 3.5GHz, with a minimum simulation step of 30μs. It supports real-time simulations of both electromagnetic transients and electromechanical transients in power systems. The simulator is compatible with the MATLAB/Simulink modeling environment and can execute, in real time, code compiled from graphical simulation models. The interface board of the real-time simulator provides 18 analog I/O ports and 64 digital I/O ports. DSP and RT-LAB are connected through digital I/O ports, and analog I/O ports are connected to oscilloscope for recording output waveforms.

Based on the topology shown in Fig 14, the VSG system model is built on the host computer. The model is compiled to generate code, which is then loaded into the real-time simulator for execution. The real-time simulation model in RT-LAB consists of two parts: the main circuit and the control system. To facilitate a comparative analysis of the experimental results, the system is controlled using both the traditional VSG control strategy and the improved VSG control strategy proposed in this paper.

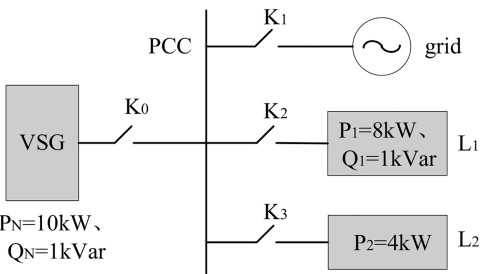

**Fig 14. VSG off-grid and on-grid system topology.**

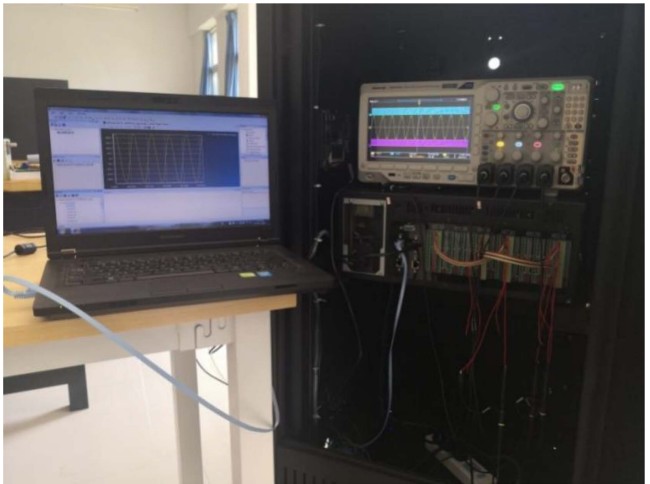

**Fig 15. Real-time simulation experiment platform based on RT-LAB.**

## Verification results of off-grid operation

When $K_1$ is open and $K_0$, $K_2$, $K_3$ are closed, the VSG operates in off-grid mode. The rated active power of VSG is 10kW, $L_1$ is a resistive inductive load (8kW/1kVar), and $L_2$ is the resistive load (4kW). The load variation sequence is summarized in Table 1, and the control strategies together with the corresponding parameter settings are listed in Table 2.

Fig 16 shows the frequency and active power waveforms of the traditional VSG for the same moment of inertia $J$ and for damping values $D$ equal to zero and to $K_p$, respectively. It can be seen that when traditional VSG control is adopted, damping plays a key role in stabilizing the inverter output power and frequency under load changes. For a fixed $J$, a larger $D$ provides greater system damping and reduces the tendency to oscillate.

**Table 1. Load variation.**

| Time(s) | Switch status | Loads(kW) | Power balance situation |
|---|---|---|---|
| $0 < t < 0.1$ | $K_0$, $K_1$, $K_2$, $K_3$ off | 0 | startup stage of VSG |
| $0.2 < t < 1$ | $K_1$, $K_3$ off, $K_0$, $K_2$ on | 8 | power of VSG is greater than load demand |
| $1 < t < 2$ | $K_1$ off, $K_0$, $K_2$, $K_3$ on | 12 | power of VSG is less than load demand |
| $2 < t < 3$ | $K_1$, $K_3$ off, $K_0$, $K_2$ on | 8 | power of VSG is greater than load demand |

**Table 2. Control strategy and control parameters.**

| Order | Control strategy | Parameters |
|---|---|---|
| 1 | Traditional VSG ontrol | $J=5$, $D=0$ |
| 2 | Traditional VSG ontrol | $J=5$, $D=K_{\mathrm{P}}$ |
| 3 | Improved VSG control | $J=5$, $D=0$, $C=0.2$, $T=0$ |
| 4 | Improved VSG control | $J=5$, $D=0$, $C=0.4$, $T=0$ |
| 5 | Improved VSG control | $J=5$, $D=0$, $C=0.4$, $T=0.02$ |

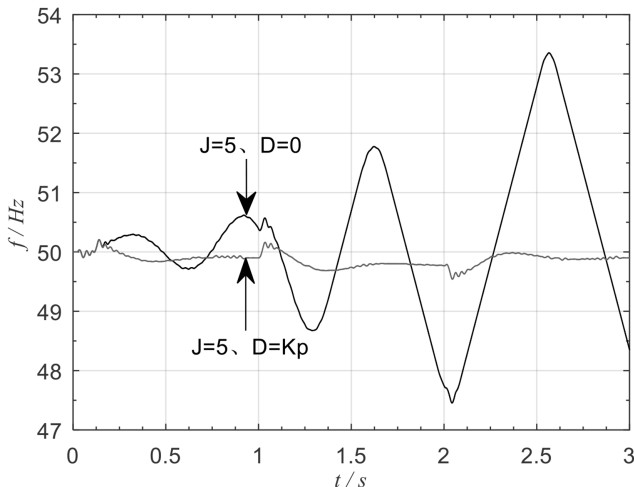

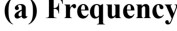

**(a) Frequency**

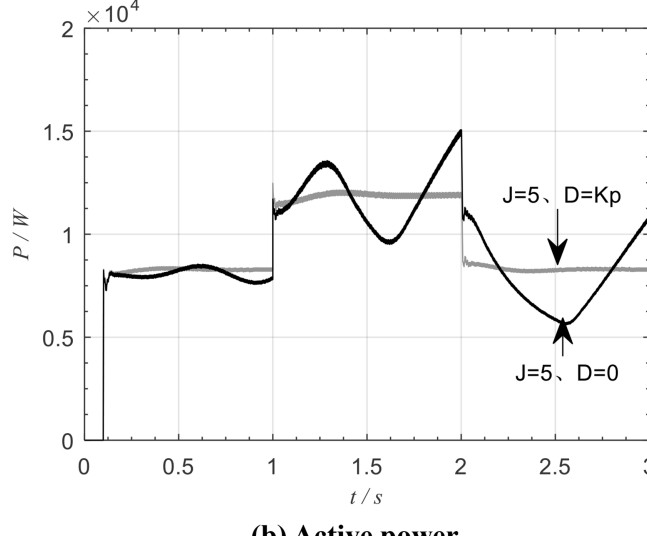

**(b) Active power**

**Fig 16. Traditional VSG frequency and active power.**

Fig 17 shows the frequency and active power waveform of the improved VSG for the same moment of inertia $J$, zero damping $D$, and different values of $C$ and $T$.

As can be seen from Fig. 17, when the load changes, a larger adjustment parameter $C$ results in smaller overshoots in both the inverter frequency and the active power. A larger inertia adjustment parameter $T$ gives the system greater inertia. The values of $C$ and $T$ do not affect the steady-state values of the active power and frequency.

### Verification results of on-grid operation

When $K_0$ and $K_1$ are closed, $K_2$ and $K_3$ are opened, VSG is connected to the grid, with a grid-dispatch command power of 10 kW. The control strategies and parameter settings are shown in Table 3.

Fig 18 shows the frequency and active-power waveforms of the traditional VSG for an inertia $J=2$ and damping $D$ values of zero and $K_P$, respectively. It can be seen that, due to the constraints imposed by the grid, the steady-state frequency of the system is 50 Hz in both cases. Without damping, the transient duration is longer, and the overshoots of the active power and frequency are also larger.

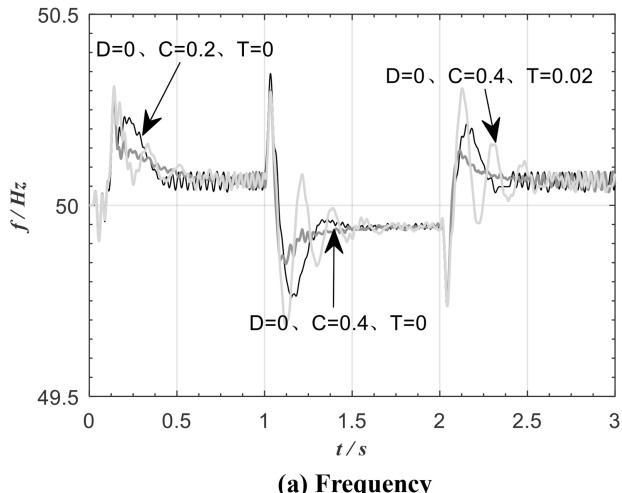

**(a) Frequency**

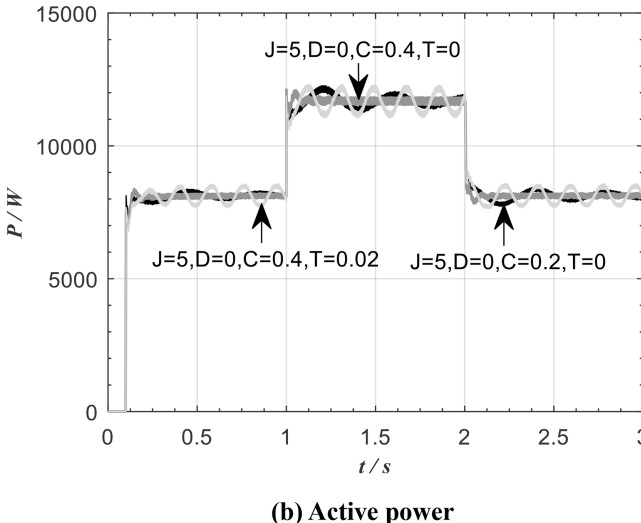

**(b) Active power**

**Fig 17. Variation of improved VSG frequency and active power.**

**Table 3. Control strategy and parameter setting.**

| Order | Control strategy | Parameters |
|---|---|---|
| 1 | Traditional VSG ontrol | $J=2$, $D=0$ |
| 2 | Traditional VSG ontrol | $J=2$, $D=K_\mathrm{P}$ |
| 3 | Improved VSG control | $J=2$, $D=0$, $C=0.2$, $T=0$ |
| 4 | Improved VSG control | $J=2$, $D=0$, $C=0.4$, $T=0$ |
| 5 | Improved VSG control | $J=2$, $D=0$, $C=0.4$, $T=0.02$ |

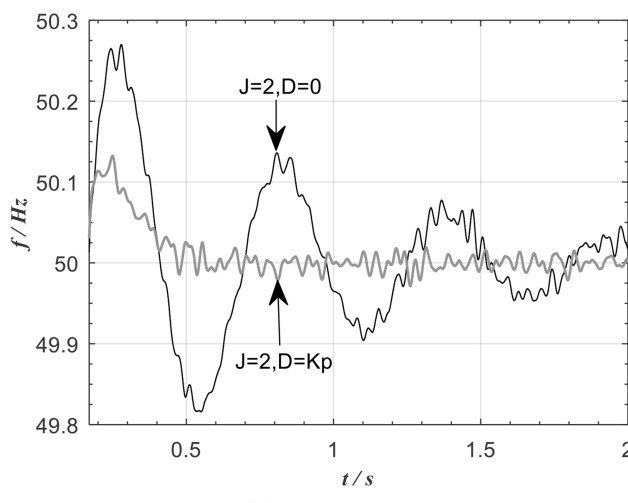

**(a) Frequency**

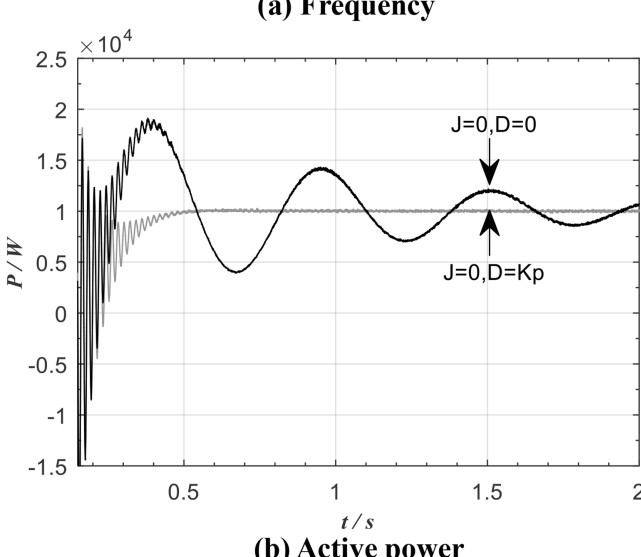

**(b) Active power**

**Fig 18. Frequency and active of traditional VSG.**

Fig 19 shows the frequency and active power waveform of the improved VSG for an inertia $J=2$, zero damping $D$, and different values of $C$ and $T$. It can be observed that, due to the introduction of the leading differential link, the active power and frequency output from the VSG stabilize after approximately 0.8s of fluctuation. A larger $C$ yields smaller overshoots

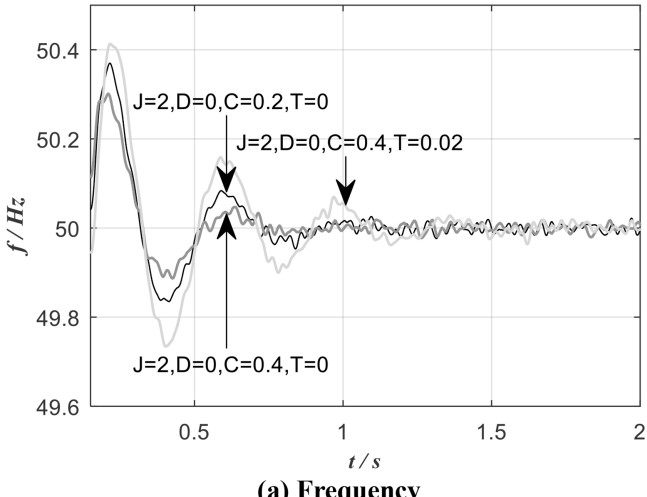

**(a) Frequency**

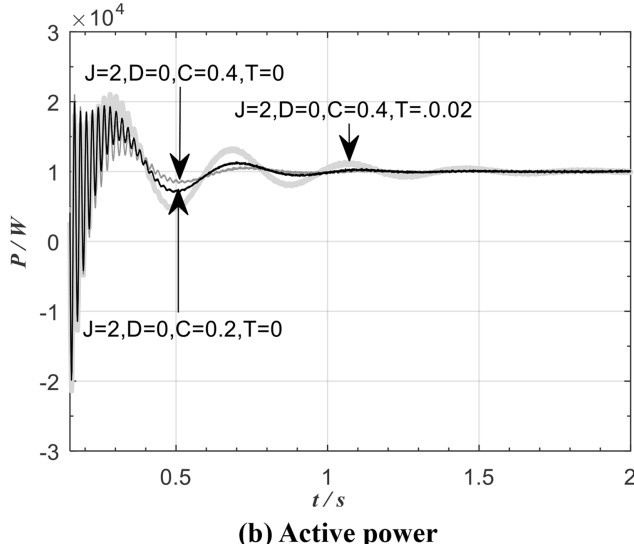

**(b) Active power**

**Fig 19. Frequency and active of the improved VSG.**

in both the active power and frequency of the VSG, while a larger *T* increases the system inertia. The experimental results above demonstrate that the leading differential link effectively suppresses frequency and active power overshoots of the inverter, and enables the decoupling control of the VSG's dynamic characteristics and steady-state performance.

## Conclusion

In view of the adjustment contradiction between the dynamic characteristics and steady-state performance of traditional VSG, an improved control strategy of VSG is proposed. The mathematical model of traditional VSG control algorithm is analyzed, and a closed-loop transfer function relating active power to frequency variations is derived. The influences of inertia and damping on active power response during transients are then investigated. By incorporating a leading differential control link and an inertia compensation link into the traditional VSG power-frequency controller, a closed-loop transfer function of active power for the improved VSG is established. The power-frequency characteristics of the improved VSG

based on the leading differential control in off-grid and on-grid states are analyzed. Adjusting the control parameters of the leading differential link proves effective in suppressing transient power oscillations, thereby achieving decoupled control of the VSG's dynamic and steady-state performance.

## Author contributions

**Conceptualization:** Weiqiang Zheng, Changwei Gao.

**Data curation:** Modan Su.

**Methodology:** Weiqiang Zheng.

**Software:** Modan Su.

**Writing – original draft:** Weiqiang Zheng.

**Writing – review & editing:** Changwei Gao.

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
