## [Decision Letter · Decision Letter 0]

10 Mar 2026

PONE-D-26-03199An improved control strategy of power-frequency characteristics of virtual synchronous generatorPLOS One

Dear Dr. Gao,

Thank you for submitting your manuscript to PLOS ONE. After careful consideration, we feel that it has merit but does not fully meet PLOS ONE’s publication criteria as it currently stands. Therefore, we invite you to submit a revised version of the manuscript that addresses the points raised during the review process.

**ACADEMIC EDITOR:** The readability of the paper needs to be further improved, the quality of the figures should be enhanced, and the descriptions of some terms should be kept consistent.

We look forward to receiving your revised manuscript.

Kind regards,

Jiapeng Liu, Ph.D.

Academic Editor

PLOS One

**Journal Requirements:**

“This study was funded by science and technology project of Liaoning Electric Power Co., Ltd. “Research on optimal control method of power frequency active support capability of PV virtual synchronous power supply (2025YF-34).”

5. Please note that funding information should not appear in any section or other areas of your manuscript. We will only publish funding information present in the Funding Statement section of the online submission form. Please remove any funding-related text from the manuscript.

6. Thank you for stating the following in the Financial Disclosure section:

“This study was funded by science and technology project of Liaoning Electric Power Co., Ltd. “Research on optimal control method of power frequency active support capability of PV virtual synchronous power supply (2025YF-34).”

We note that you received funding from a commercial source: Liaoning Electric Power Co., Ltd

7. We note that one or more of the authors are employed by a commercial company: Yingkou Power Supply Company

8. We note that your Data Availability Statement is currently as follows:

“All relevant data are within the manuscript and its Supporting Information files.”

9. PLOS requires an ORCID iD for the corresponding author in Editorial Manager on papers submitted after December 6th, 2016. Please ensure that you have an ORCID iD and that it is validated in Editorial Manager. To do this, go to ‘Update my Information’ (in the upper left-hand corner of the main menu), and click on the Fetch/Validate link next to the ORCID field. This will take you to the ORCID site and allow you to create a new iD or authenticate a pre-existing iD in Editorial Manager.

**Additional Editor Comments:**

The paper needs to be strengthened in the following aspects:

1. The paper should provide the tuning method of control parameters. From the perspective of controller design, the proposed method relies too heavily on the model. When the model parameter identification is inaccurate, can the effectiveness of the algorithm be guaranteed?

2. The stability analysis needs to be supplemented. According to the experimental data, obvious chattering appears in the control error. What is the cause of this phenomenon?

3. lease supplement the diagrams of experimental equipment and describe the experimental setup in detail.

Reviewers' comments:

Reviewer's Responses to Questions

**Comments to the Author**

1. Is the manuscript technically sound, and do the data support the conclusions?

Reviewer #1: Yes

Reviewer #2: Yes

2. Has the statistical analysis been performed appropriately and rigorously? 

Reviewer #1: Yes

Reviewer #2: Yes

3. Have the authors made all data underlying the findings in their manuscript fully available?

Reviewer #1: Yes

Reviewer #2: Yes

4. Is the manuscript presented in an intelligible fashion and written in standard English?

Reviewer #1: Yes

Reviewer #2: Yes

5. Review Comments to the Author

Reviewer #1: This paper proposes an improved control strategy for VSG to address the regulation contradiction between virtual inertia and damping. The topic is interesting. However, the contributions of this work appear to lack novelty. The reviewer offers the following comments for consideration:

1. In the introduction, some relevant studies on power oscillation damping methods for VSGs from the last five years are missing. For instance, virtual impedance control, additional damping control, and mutual damping control can also reduce control overshoots without affecting steady-state performance.

2. The analysis in grid-connected mode assumes a strong grid with infinite capacity ( ωg = ω0 ). In practical applications, grid strength may vary and can affect the system's dynamic response. The authors are encouraged to analyze and evaluate the robustness of the proposed control strategy under varying grid strength conditions.

3. The manuscript only considers a single VSG. It would be valuable to investigate the performance and scalability of the proposed control strategy when applied to systems with multiple VSGs operating in parallel.

4. The superiority of the proposed control method requires further validation. Specifically, its performance should be compared with state-of-the-art power oscillation damping methods to demonstrate its advantages, at least by providing a comparative analysis with existing adaptive methods (e.g., VSG with adaptively tuned J and D).

Reviewer #2: The author proposed a method to supress power oscillation of virtual synchronous generator (VSG) in the transition process. The mathematical model of VSG, control algorithm and closed-loop transfer function of active power to frequency fluctuation are developed. The simulation results show that the proposed method is good to supress the power oscillation.

Hence, this manuscript can be accepted.

6. PLOS authors have the option to publish the peer review history of their article (what does this mean?). If published, this will include your full peer review and any attached files.

Reviewer #1: **Yes:** Zhengrong Xiang

Reviewer #2: **Yes:** VIJAYAN VELAPPAN

---

## [Author Response · Author response to Decision Letter 1]

30 Mar 2026

Thanks to the valuable suggestions from the editor and reviewers, the author has made revisions to the manuscript according to the reviewers' comments.

---

## [Decision Letter · Decision Letter 1]

15 Apr 2026

PONE-D-26-03199R1An improved control strategy of power-frequency characteristics of virtual synchronous generatorPLOS One

Dear Dr. Gao,

Thank you for submitting your manuscript to PLOS ONE. After careful consideration, we feel that it has merit but does not fully meet PLOS ONE’s publication criteria as it currently stands. Therefore, we invite you to submit a revised version of the manuscript that addresses the points raised during the review process.

**ACADEMIC EDITOR:** The readability of the manuscript needs to be further improved to enhance its appeal.==============================

We look forward to receiving your revised manuscript.

Kind regards,

Jiapeng Liu, Ph.D.

Academic Editor

PLOS One

Journal Requirements:

Additional Editor Comments:

The paper still has certain deficiencies in the analysis of research status and the description of experimental equipment. It is suggested to supplement the analysis of research status in the past three years and provide a detailed introduction to the experimental equipment.

Reviewers' comments:

Reviewer's Responses to Questions

**Comments to the Author**

1. If the authors have adequately addressed your comments raised in a previous round of review and you feel that this manuscript is now acceptable for publication, you may indicate that here to bypass the “Comments to the Author” section, enter your conflict of interest statement in the “Confidential to Editor” section, and submit your "Accept" recommendation.

Reviewer #1: All comments have been addressed

Reviewer #2: All comments have been addressed

2. Is the manuscript technically sound, and do the data support the conclusions?

Reviewer #1: Yes

Reviewer #2: Yes

3. Has the statistical analysis been performed appropriately and rigorously? 

Reviewer #1: Yes

Reviewer #2: Yes

4. Have the authors made all data underlying the findings in their manuscript fully available?

Reviewer #1: Yes

Reviewer #2: Yes

5. Is the manuscript presented in an intelligible fashion and written in standard English?

Reviewer #1: Yes

Reviewer #2: Yes

6. Review Comments to the Author

Reviewer #1: All my concerns have been addressed satisfactorily.

Thus, this paper can be acceptable.

Reviewer #2: The authors made sufficient corrections as per the reviewer comments in this revised manuscript. Hence, it can be accepted.

7. PLOS authors have the option to publish the peer review history of their article (what does this mean?). If published, this will include your full peer review and any attached files.

Reviewer #1: No

Reviewer #2: **Yes:** VIJAYAN VELAPPAN

---

## [Author Response · Author response to Decision Letter 2]

19 Apr 2026

Dear Editors:

Thank you for your comments concerning our manuscript entitled “An improved control strategy of power-frequency characteristics of virtual synchronous generator”. We have studied comments carefully and have made correction which we hope meet with approval. The correction in the paper and the responds to your comments are as following:

Responds to the comments:

Academic editor Comments:

The readability of the manuscript needs to be further improved to enhance its appeal.

Author Response:

According to the editor's requirement, the author polished the entire manuscript to enhance its readability. The revised content can be found in the blue fields of the revised manuscript.

Additional Editor Comments:

The paper still has certain deficiencies in the analysis of research status and the description of experimental equipment. It is suggested to supplement the analysis of research status in the past three years and provide a detailed introduction to the experimental equipment.

Author Response:

In the revised manuscript, the author updated and replaced references 1, 5, 7, and added references 18, 21, 29, 30, 31. In the introduction, the author also conducted supplementary analysis of the research status in the past three years. The modified and supplementary content can be found in the red fields of the introduction and references.

According to the editor's requirement, the author provides additional descriptions of the experimental equipment in the revised manuscript. The supplementary content can be found in the red fields of "Experiment verification".

---

## [Editor Report · Decision Letter 2]

21 Apr 2026

An improved control strategy of power-frequency characteristics of virtual synchronous generator

PONE-D-26-03199R2

Dear Dr. Gao,

We’re pleased to inform you that your manuscript has been judged scientifically suitable for publication and will be formally accepted for publication once it meets all outstanding technical requirements.

Kind regards,

Jiapeng Liu, Ph.D.

Academic Editor

PLOS One
---

## [Editor Report · Acceptance letter]

PONE-D-26-03199R2

PLOS One

Dear Dr. Gao,

I'm pleased to inform you that your manuscript has been deemed suitable for publication in PLOS One. Congratulations! Your manuscript is now being handed over to our production team.

Kind regards,

on behalf of

Dr. Jiapeng Liu

Academic Editor

PLOS One